# Determinants predicting the electronic medical record adoption in healthcare: A SEM-Artificial Neural Network approach

**Amina Almarzouqi[1], Ahmad Aburayya[2], Said A. Salloum[3]***

**1** Department of Health Service Administration, College of Health Sciences, University of Sharjah, Sharjah, UAE, **2** Doctor of Quality & Operation Management, Quality & Corporate Development Office, Dubai Health Authority, Dubai, UAE, **3** School of Science, Engineering, and Environment, University of Salford, Manchester, United Kingdom

\* salloum78@live.com

**Data Availability Statement:** There are legal restrictions on sharing de-identified data. The data are not publicly available due to the Dubai Health Organization's policy in protecting its employees'

## Abstract

An Electronic Medical Record (EMR) has the capability of promoting knowledge and awareness regarding healthcare in both healthcare providers and patients to enhance interconnectivity within various government bodies, and quality healthcare services. This study aims at investigating aspects that predict and explain an EMR system adoption in the healthcare system in the UAE through an integrated approach of the Unified Theory of Acceptance and Use of Technology (UTAUT), and Technology Acceptance Model (TAM) using various external factors. The collection of data was through a cross-section design and survey questionnaires as the tool for data collection among 259 participants from 15 healthcare facilities in Dubai. The study further utilised the Artificial Neural Networks (ANN) algorithm and the Partial Least Squares Structural Equation Modeling (PLS-SEM) in the analysis of the data collected. The study's data proved that the intention of using an EMR system was the most influential and predictor of the actual use of the system. It was also found that TAM construct was directly influenced by anxiety, innovativeness, self-efficacy, and trust. The behavioural intention of an individual regarding EMR was also proved to positively influence the use of an EMR system. This study proves to be useful practically by providing healthcare decision-makers with a guide on factors to consider and what to avoid when implementing strategies and policies.

## 1. Introduction

Over the years, Health Information systems are seen to have changed from the initial use of paper to store records to using electronic forms in most healthcare institutions found in developed countries. According to [1, 2], the adoption of an EMR system is to provide better and efficient healthcare services to patients. The preference towards EMR can be attributed to the system being connected with the clinical decision support system (DSS), which is significant in providing support in decision-making for all healthcare providers. In addition, as put by [3],

information. Data would be provided upon reasonable request from the Quality and Excellence Office at PHA (contact via: QEO-Excellence@dha. gov.ae).

**Funding:** Unfunded studies The author(s) received no specific funding for this work.

**Competing interests:** NO authors have competing interests. The authors have declared that no competing interests exist.

EMR helps in making a fast and efficient decision in areas, such as data analysis, diagnosis, billing, and lab results.

Health Information System (HIS), according to [4], is any system that is capable of transmitting, managing, storing, and capturing patients' medical records and information related to activities undertaken within an organisation in the health sector. Moreover, as put by [5], HIS's main objective is the provision of healthcare services to patients. However, with the introduction of EMR in HIS, the initial doctor-patient relationship is now at an advanced level consisting of a larger healthcare team system that guarantees the provision of better quality healthcare services to patients [4].

Healthcare institutions in developed countries have been using the paper-based system as their main method of recording patient data [6]. However, [7] study identifies challenges related to the paper-based recording system that include data disintegration, incomplete data, illegibility of data, and the recorded data being ambiguous. In addition, as stated by [8, 9], recording and storing of patients' data on paper prevents the flow of information which affects the healthcare system's competence hence the need to adopt the EMR system as a solution to such challenges. Moreover, the EMR system is advantageous in facilitating an effective way of retrieving information, making it possible to conduct validity tests regarding the quality of data, enhancing better decision-making in the choice of treatment to a condition, and also helpful for research purposes [4, 5, 10] on the significance of EMR, it is possible for patients suffering from chronic diseases to be admitted once hence doing away with cases of readmissions of such patients whenever they need health services.

The development of an EMR system is through four sequential stages that are EMR, electronic patient records, computerised medical records, and automated medical records [4]. According to [11], the application of EMR in the HIS is specifically for handling related practical applications, making medical decisions, taking care of patient's treatment, and as a support to clinical records. They can further be categorised into; handling of healthcare electronic connectivity and communications; order-management such as the use of templates; management of health records such as radiology and laboratory tests; and recording of patient information [12].

More on the use of EMR is by [13], whose findings indicate EMR's significance in the improvement of medical quality, lowering medical costs, facilitating effective medical care delivery, enhancing the safety of patients, and reducing risks related to adverse drug effects in inpatient and emergency settings. An illustration of the above is seen in a health facility in a developed country whereby EMR reduced laboratory cost, transcription time, and length of stay with a benefit of over five years of US$ 613,681 [14]. In addition, in the USA, the introduction of an EMR system in the outpatient setting resulted in a low cost of spending and attaining a higher revenue of approximately US$ 952,000 as compared to the previous year in the absence of an EMR system [15].

To facilitate the successful adoption of healthcare technologies, it is essential to be knowledgeable in various factors affecting the application of such technology [16]. According to a study conducted in the Gulf region, specifically in the UAE concerning the adoption of Electronic Medical Records, [15, 16] show that there are essential gaps in expounding and predicting the advocacy of the EMR wholesomely. Furthermore, the insightful perspective of the literature present at the moment shows that there are specific elements that were studied that give a prediction of various levels of adoption of the EMR. Application of various methods or models of acceptance were used, which include UTAUT and TAM. In as much as there are models used to analyse the acceptance of the EMR methods, there lacks a model that is comprehensive enough to enable coverage of distinctive and wild scope trends in the medical care field in the UAE. Thus, the current research aims to examine the expounding elements and

give predictions of the Electronic Medical Records acquisition in the medical care field. The current research study aims at achieving the following at its completion: To start with, the research tries to understand previous effects brought about by the adoption of EMR, either directly or indirectly. For this to be achieved, the study has to undertake the use of an innovative and integrated research model to facilitate the identification of the determinants that lead to EMR adoption. Therefore, the study employed a conceptual model able to interconnect the UTAUT theory with the TAM acceptance model associated with external factors identified in the study, highlighting the certainty and significance of the findings. In addition, the research study is involved with the evaluation of the success related to an EMR-adoption in a healthcare setup. Certainly, embracing an EMR system is not only beneficial to the direct users but the community at large, such as sharing of information between healthcare providers and individuals within different setups in a community aimed at aligning their attitudes and behaviours towards better health. Further, the current research has employed various external variables to particularly identify the significance of external features associated with EMR within the healthcare setting. Unlike previous studies such as [2, 4, 17], which focused on confidentiality, usability, and availability as their external factors, this study's external factors related to EMR revolve around trust, innovativeness, self-efficacy, and anxiety. Finally, this study employs a different methodology to analyse its data by the use of an ANN analysis tool that is also identified as the best tool for prediction in adopting health technology. This is also the first attempt in the medical field with the help of the SEM-ANN approach and an integrated model aimed at filling the identified gaps in the current research study.

In developed countries, the healthcare sector has consistently tried to maintain an EMR system with the aim of cost reduction for healthcare services and improving the quality of such services. According to [2], the USA in an aim to facilitate the adoption of an EMR system in all hospitals by 2014, provided the healthcare sector with a $1.2 billion grant. This introduction is expected to improve the exchange of Health Information and manage a Nationwide Health Information Network (NHIN) able to enhance an interoperable and secure infrastructure to facilitate healthcare stakeholders into making better decisions regarding healthcare [18, 19]. However, the transition towards an EMR system in the US healthcare sector has not been embraced in the near past as it was expected. According to [20], there was a legislative order signed in 2009 requesting health institutions to adopt an EMR system in a move to reduce mortality rates, prescription errors, and medical errors. However, the pressure accompanying the order was overwhelming, especially for the participation from the top-level management indirectly affecting the transition process.

Adoption of an EMR system in the healthcare setting is one of the major concerns of healthcare researchers in countries, such as Bangladesh, Taiwan, the Netherlands, and the USA, with most of these researchers using surveys as a tool of collecting data. According to research conducted by [2, 21], approximately 78 elements are influencing the adoption of an EMR system. The elements are further put into 8 groups that include technical, financial, legal, environmental, organisational, psychological, behavioural, and individual factors. Further, [22] study identifies 6 factors critical for the adoption of an EMR system. This includes; expert support, communication between users, technical support, interoperability, workflow impact, and user attitude towards health information systems. On the contrary, [23] research shows an indirect influence between EMR-adoption by physicians and factors such as data security, misfit, trust, self-efficacy, and anxiety.

Further research by [24], consisting of 1075 physicians, proves that the significance and satisfaction associated with EMR positively influence the use of EMR purposely for outpatients. A similar study by [25] shows a direct correlation between EMR assimilation and community identity for healthcare providers. In addition, according to [12], the adoption of an EMR

system, especially in a small setup is significantly influenced by financial factors. A different study by [26] posits that the trend in using EMR systems can be affected by performance expectancy and demographics. In addition, according to [13], factors such as computer sophistication, training, and age were influential towards the adoption of an EMR system.

Based on 142 research studies conducted between 2010 and 2022 on the acceptance of incorporating healthcare and technology, [16] conclude that the improvement in TAM is foremost in studies that focus on the adoption of technology in the medical field. In addition, some of the studies focus on TAM's integration with other models of technology such as UTAUT that, according to [16], was greatly used by some of studies in exploring technology acceptance in healthcare. Further, more studies on the TAM model focus on its main features that are perceived ease of use and perceived usefulness in the exploration of technology acceptance in healthcare, such as [27, 28]. Moreover, a set of different studies from the above prove that the use of technology due to behavioural intention is the most utilised feature in the exploration of technology acceptance in the healthcare setting [16]. Also, studies by [2, 4, 16, 29–33] show that, other than UTAUT and TAM, trust, anxiety, self-efficacy, and innovativeness, are also useful in studying the relationship technology acceptance in healthcare. Therefore, our research seeks to answer the following research questions:

- What are the factors that predict and affect EMR adoption in healthcare?

- How do the most widely used external factors of TAM & UTAUT affect the EMR adoption in healthcare?

## 2. Research model and hypotheses-development

This study applied the use of the UTAUT model and TAM as the study's main tool in investigating factors influencing EMR adoption in healthcare institutions in Dubai [34]. Moreover, the UTAUT model, according to [35], is 70 per cent efficient hence widely utilised in areas related to the EMR systems. In addition, guided by previous studies that applied the UTAUT model in healthcare, they chose to use the most current constructs of the model in undertaking the study which includes facilitating conditions, performance expectancy, effort expectancy, and social influence. The above-mentioned constructs are aspects that influence behavioural intention among individuals hence affecting the EMR system adoption. However, the TAM model is also crucial as part of the development model. This is because the TAM model has a direct effect on EMR adoption as it is connected to behavioural intention in two ways that are, perceived usefulness, and ease of use. According to [2, 4, 16, 29–33], the two ways in which TAM model is connected to behavioural intention are critical factors in investigating EMR's adoption in healthcare and might be affected by the study's external factors (see Fig 1).

### 2.1 TAM external variables

As stated by [36], trust, which is also one of the external factors affecting TAM, is a person's belief in being safe and secure in using a particular technology. Trust can be viewed as an influence of loyalty, use, and acceptance regarding the adoption of technology in healthcare settings hence affecting EMR adoption either positively or negatively [29, 37]. Further, [36, 38] study, while analysing past research regarding technology acceptance in the healthcare system, conclude that factors related to trust have an influential role towards perceived usefulness (PU). Trust is also an influence towards the ease of use as it reduces individuals' control, monitor, and the need to understand technology hence reducing cognitive effort and enhancing digital transactions and communications. Therefore, the higher the level of trust, the higher the perceived ease of use (PEU) with the reverse also being applicable [29, 37, 38]. The above proves a

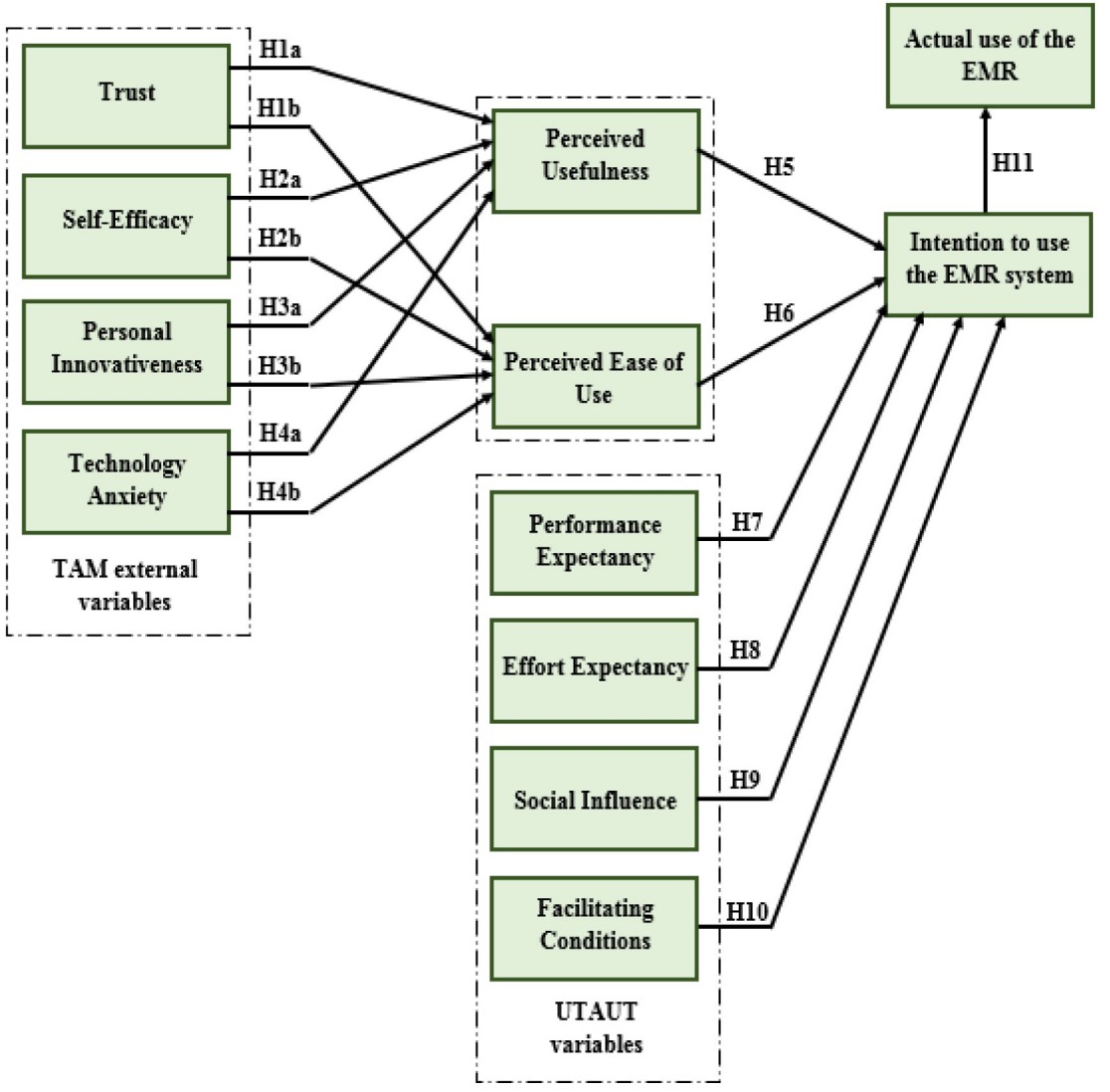

**Fig 1. Research model and hypotheses.**

significant relationship between trust and perceived ease of use which helps in understanding the acceptance of technology in healthcare hence the development of the following hypotheses:

**H1a:** PU is positively influenced by trust in the adoption of an EMR system.

**H1b:** PEU is positively influenced by trust in the adoption of an EMR system.

According to [39], self-efficacy is the trust in a person to be able to do a particular activity using a computer. Similarly, [40] theory on self-efficacy proves that individuals that have high self-efficacy experience a positive effect as compared to individuals with low self-efficacy. Also, according to [23, 41], self-efficacy has a positive impact on perceived usefulness (PU) and perceived ease of use (PEU) in the acceptance of technology in healthcare hence the following hypotheses:

**H2a**: PU is directly influenced by self-efficacy in the adoption of an EMR system.

**H2b**: PEU is positively affected by self-efficacy in the adoption of an EMR system.

A study on personal innovativeness by [42] shows a direct relationship with a user's readiness in using technological inventions once they are available. Further, personal innovativeness is connected to an individual's perception of technology and confidence whereby a higher level of personal innovativeness, translates to higher levels of confidence in an individual's actions. [42] also point out that individuals with high innovativeness are likely to perceive technology positively. With the fact that users make different choices regarding technological adoptions, personal innovativeness is marked as one of the choices for individuals hence directly affecting how they perceive technology. TAM theory tries to explain the above whereby personal innovativeness, perceived ease of use (PEU) and perceived usefulness (PU) have a positive connection [6, 42–44] which helps in the formulation of the following hypotheses;

**H3a:** PU is influenced by personal innovativeness in the adoption of an EMR system.

**H3b:** PEU is influenced by personal innovativeness in the adoption of an EMR system.

Another factor affecting TAM is Technology Anxiety. According to [45], technology anxiety is the fear brought about by the probability of having to use technology. [46, 47] further prove a significant connection between anxiety and both PU and PEU regarding the acceptance and use of technology in healthcare hence developing the following hypotheses:

**H4a:** PU is influenced by technology anxiety in the adoption of an EMR system.

**H4b**: PEU is influenced by technology anxiety in the adoption of an EMR system.

## 2.2 TAM, UTAUT, BI & EMR adoption

According to [48], perceived usefulness and ease of use play critical roles in studying the adoption of technology in the healthcare system. In addition, [4] study state that PU and PEU create a BI and a positive attitude toward the acceptance and use of EMR. Further, PU and PEU have proved the existence of a positive influence on healthcare providers' use and adoption of an EMR system. [49] study state that healthcare providers are not used to changing their behaviour due to job specificity. Therefore, it is the perceived usefulness of EMR that should be updated with an aim of convincing healthcare providers to adopt the system [26].

According to findings from [50] study, the PEU by physicians has a direct influence on their PU towards mobile EMR hence both influencing the use of EMR. Similarly, [51] conclude the existence of a relationship between PEU and PU and their role together in influencing the use of evidence adaptive clinical decision support systems. With the help of the integrated model, psychosocial model, extended TAM, and TAM, [52] performed an investigation on the adoption of EMR by healthcare providers. According to their findings, PEU and PU are interconnected, and that both play a critical part in the use of an EMR system by healthcare providers [52] which is also confirmed by [26, 53]. Further, [54] proves the existence of a positive correlation between behavioural intention and both PU and PEU hence the following hypotheses;

**H5:** The PU of EMR systems by Healthcare providers has a significant influence on their BI to use EMR.

**H6:** The PEU of EMR systems by Healthcare providers has a direct influence on their BI to use EMR.

According to the current research, a user's BI towards the use of EMR is influenced by factors such as Performance Expectancy, Effort Expectancy, Social Influence, and Facilitating Conditions [2]. Performance Expectancy, according to [55, 56] are greatly influential

regarding BI towards the use of new technology in enhancing the health information system in general [52, 57]. According to [9, 58, 59], Effort Expectancy is significant in positively influencing users to adopt new technology such as eHealth services and the electronic medical record system. Further, the influence and relationship that SI has on BI in the adoption of new health information systems are confirmed by [2, 60, 61]. Also, according to [2, 62], Facilitating Conditions are crucial in influencing behavioural intention towards accepting new technologies with infrastructural support. The above findings formulate the following hypotheses;

**H7:** Performance expectancy is a facilitating factor towards a healthcare provider's intention to adopt an EMR system.

**H8:** Effort expectancy is a facilitating factor towards a healthcare provider's intention to adopt an EMR system.

**H9:** social influence is a facilitating factor towards a healthcare provider's intention to adopt an EMR system.

**H10:** facilitating conditions are highly influential towards a healthcare provider's intention to adopt an EMR system.

Behavioural intention, as put by [63, 64] is the aim of learning a particular behaviour in the future which can therefore act as a predictor of a person's future adoption of new technology in any particular field. Also, according to [34], behavioural intention influences the prediction of adopting a specific technological system. For instance, [65] prove that it is possible to detect the likelihood of an individual adopting a new technology through their intention to use it. Therefore, it is precise that Behavioural Influence presents a significant influence towards the adoption of an EMR system by healthcare providers [2, 17, 66, 67]. This is because the intention of using new technology such as EMR is most likely to lead to the actual use hence the following hypotheses;

**H11:** Behavioural intention towards the use of an EMR system is capable of influencing the actual use of the system.

## 3. Research methodology

This study employed the use of a cross-sectional design as a deductive strategy for the formulation of hypotheses. A drop-off questionnaire was applied as the main instrument for data collection that was self-administered among healthcare providers in the UAE. The research study was held at a healthcare facility based in the City of Dubai with the participation of 3 hospitals and 12 primary healthcare clinics for 3 months. On the first page of the survey, there was an information sheet and a consent form. Respondents were able to leave at any moment without justification, and no personal identification was required to protect the privacy of the data. Respondents were not compensated in any way for taking part in the survey. The ethical permission letter for this study was issued by University Students Research Evaluation Committee (USREC01-04/DBA/2017), allowing it to conduct surveys in the DHA's facilities. However, the current study adhered to DUBAI SCIENTIFIC RESEARCH ETHICS COMMITTEE (DSREC-SR-02/2017_01).

The data collected was from various healthcare workers who were grouped into administrative staff and clinical staff from the selected 15 healthcare institutions. With this study taking the nature of empirical research regarding healthcare service management, it would be effective to opt for the target population as the main source of data [3]. According to [68], in most cases, individuals familiar with a particular field have a significant level of knowledge on that

field, hence of help in gaining information such as customer satisfaction and the degree of service quality offered in their field of work.

Moreover, the sampling technique chosen for the current study is non-probability through the implementation of a convenience sampling strategy, which was highly influenced by the healthcare policy found in Dubai that ensures the security and confidentiality of staff information. The choice of the sampling technique was further influenced by the specific policies of the chosen healthcare centres and the fact that convenience sampling is suitable for the access of huge samples, cost-effective, and time-friendly [69]. From the 496 questionnaires distributed to the 15 healthcare centres, the study recorded a response rate of 52.2 per cent with 259 usable questionnaires.

Having established this endorsement, healthcare facilities visits were directed to ensure the researchers and affiliates of hospitals and clinics administration were properly accustomed and reduce or mitigate the risk of non-respondent bias. At that time, the researchers explained the aims of the research and its subject to these staff members. Researchers distributed and collected the questionnaire by hand and requested help from certain medical and nursing directors at those facilities. Visiting the defendants and delivering the forms in person meant the scholar could instill confidence in the study and provide reassurance to the accused. During this interaction procedure, researchers responded to any questions of a practical nature and elucidated any points raised by the plaintiffs. More significantly, this allowed the researchers to encourage respondents that the review would only be used for hypothetical purposes and that their anonymity was guaranteed. Respondents were also informed that they, as workers, were likely to be beneficiaries of the study's results in the long term. In cases where managers were too busy to meet the researchers during these visits, questionnaires were left to be completed in the following days. In essence, healthcare facility managers were notified that a confidential box was set up inside each department for participants to deposit their completed surveys. The researchers had unique access to these lockable boxes to emphasize the dedication to secrecy.

Furthermore, the researchers used their personal and social relationships to facilitate the distribution of the questionnaires. For instance, help was sought from a number of the researchers' friends who work in the Dubai hospitals and clinics targeted in this study. These people could thus assist with distributing and collecting the questionnaires. Whenever possible, telephone calls to the hospital and clinic managers and supervisors were also conducted to encourage participation and remind respondents to complete their questionnaires.

To facilitate the current empirical study and test hypotheses, it was crucial for the study to adequately establish tools to use in measuring theoretical constructs regarding EMR adoption, behavioural intention, UTAUT, TAM external variables, and TAM that also requires consideration while dealing with the conceptual domain of the constructs. Measuring of the constructs for this particular study was based on a 5-point Likert scale, with extreme agreeing and disagreeing points. Constructs on the adoption of an EMR system were measured using 3 items as advised by [70], while constructs related to behavioural intention were measured using 3 items as advised by [71] study. Further, the UTAUT constructs were measured using the 14-item UTAUT scale developed by [2]. The two constructs representing TAM were also measured using [34] 7 items, while using 11 items from [72] to measure the four constructs in TAM's external variables. The study further employed the PLS-SEM approach in evaluating the study's theoretical model which proved significant in enhancing the results' accuracy [73] and ANN algorithm in evaluating independent variables taking the role of a complementary multi-analytical approach regarding electronic medical records.

## 3.1 Artificial intelligence analysis

The Partial Least Squares (PLS) reversion is applied in elaborate and chronological steps, non-exclusive, and numerous linear reversion modeling. The regression modeling tool is applied

based on the predictor variables such as the Xs concerning the Y variable using the equation below;

$$Y = b_0 + b_1X_1 + b_2X_2 + \ldots + b_pX_p \tag{1}$$

The modeling techniques were created to satisfy both the divergent and the convergent validity and meet the canonical correlation assumptions. The assumptions made and the methods in the modeling and the imposed restrictions include;

i. The underlying factors on the X and Y variables are obtained from the X'X and Y'Y matrices. And they are not derived from the cross-product mediums that involve both the X and Y variable quantity.

ii. The model's prediction functions cannot go beyond the minimum number of X and Y functionality variables ([74], p. 234). The partial least square regression model is one of the least restraining multivariate modeling tools. Along with the study done by SatfSoft (2013), it is clear that the prediction functions and PLS regression are represented using the factors that are derived from the Y'XX'Y matrix.

The amount of these estimate functions derived from the matrix can be obtained characteristically, which will surpass the least number of the X and Y variables. The majority of the studies conducted in the study of the information systems and the EMR systems are dependent on the tools that include partial least squares (PLS) and structural equation modeling (SEM), regression which is vital in testing the causal relationships [75–78]. Despite the universal acceptance of the application of this method, there is a possibility that it can lead to the over-simplification of the calculation's complexities during the decision-making process [79, 80]. With the models being considered in the artificial intelligence technique, the application of ANN has been widely used recently in information systems. It has been used as the technology acceptance and adoption (i.e., [75–77, 81–84]), this is mainly because it can outperform other modeling tools like the lined modeling, regression, and the PLS-SEM. It is majorly a result of its ability to fully identify both the validity prediction and the linear relationships that are coupled with the fast learning and accuracy predictions, the high prediction accuracy and the accuracy in the validity prediction, exerts the dominance in reliability over the PLS-SEM and regression, ANN is considered as less restrictive when applied–with limited mandate on the factor loadings, assumption normality, homoscedasticity, linearity, sample size, sample error and the lack of power [5, 78, 80], fault tolerance–ANN, however, is a model that can accommodate all the functions samples with an ore individual differences [85] and it also has a considerably good capability generalization–it is robust against the missing data or noisy data [86]. These considered attractive functions of the model are among the contributing factors to its versatility in application. It indicates that it is very insulated from the flaws found in the statistical application and the myths identifiable to the traditional modeling tools. Various artificial intelligence scholars such as [87] have posited that other experts need to use a variety of other modeling methods to predict the actual use of the EMR or the diagnosis of the variables in the functions.

This report responds to the call to use the ANN model because it is genuinely unique and superior compared to other modeling tools such as PLS-SEM and regression modeling techniques while conducting the estimations with a high level of results accuracy. The overall capability of the ANN modeling tool is efficient in modeling the complex interactions having flexible non-linear response values, which give it a superior prediction power to the other traditional modeling tools [88, 89]. Besides the fact that the ANN modeling technique can help detect the non-linear practical relations concealed in the given statistics, the modeling tool can

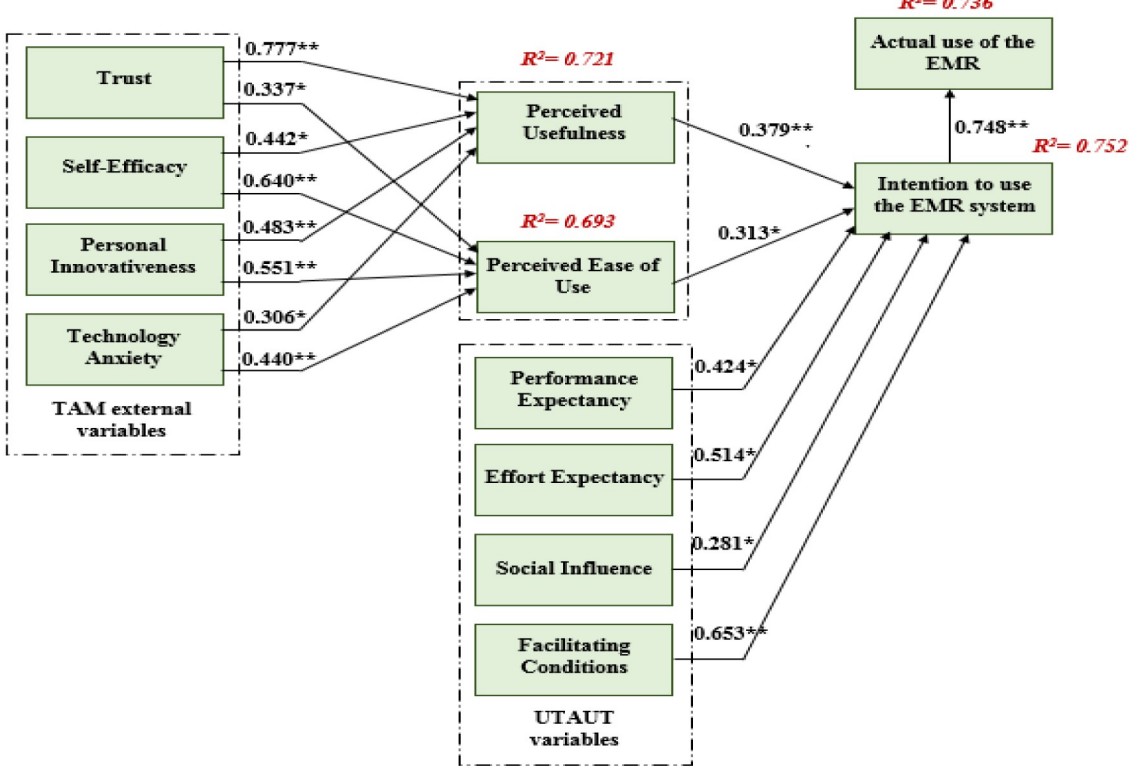

**Fig 2. Path coefficient of the model (significant at $p^{**} < = 0.01$, $p^* < 0.05$).**

use the identified relationships to the new set of data [90]. The electronic medical record adoption in healthcare is among the highly complex problems that include; ANN is considerably the most suitable modeling tool that is efficient in testing these variables. More subtly, the ANN modeling tool constitutes a data processing system that includes positive numbers of the data processing units that include neurons and the cells incorporated in the mathematical functions and directly connect to the weighted links [91]. The ANNs modeling tool includes a class of knowledge discovery which is important in solving classification and clustering, predictions, and estimation activities [79]. The data processing units are systematized in layers. The layer input provides the input information; other concealed layers will produce the information through the neutral system, and the production layer of the data gives the results [92]. ANN modeling system can help forecast both the asymmetric and symmetric relationships with an accuracy of 100% with no multivariate mandate assumptions. The multivariate assumptions are normality, linearity, and homoscedasticity. In addition, the strength of the modeling tool and the artificial intelligence method may play a role in overcoming the issue of the common method of electronic medical record adoption and bias training because the research models can be replicated and each of the cases tested for data exactness in the estimate done.

It is on this foundation that this study involves the combinations of the PLS package, such as the Partial-Least-Squares Regression (PLSR) and the ANN that includes the neural network to help predict the electronic medical record adoption in the healthcare. In the process, the PLS regression is adopted to help understand the factor structuring and help create an understanding in explaining the data modification of the predictor variable quantity on the reply variable. The application of the ANN is applied in the simulation and modeling as well as

testing the research accuracy that follows the adopted procedures before this study [75–77, 81–84]. The ANN modeling process used the multi-layer perceptron with the 3:2 concealed nodes and the robust backpropagation with the mass backtracking algorithm. The logistic role was applied in this process as a form of activation function for the output layer and the hidden function; the sum square error (SSE) was also applied to help in the differentiable error function. The process training data used was 70% and 30% for testing the procedure. The synaptic weights of the process participation nodes on both the secreted and the production nodes as illustrated in Figs 3–6 below. The main objective of the process is to help in the depreciation of the mistake till the ANN learns via training or learning procedure. In the training procedure, the arbitrary synaptic masses were given to the networks, and the goal was to change them to help obtain slight errors. The comprehensive training course needed three hundred and seventy-two steps until the complete limited derivatives of the modeling mistake functions obtained were lesser than the 0.010. The ANN modeling technique is difficult to apply; for instance, visually, the model suffers from the issue is serious clutter, which makes the interaction weights mostly smaller [93].

Furthermore, the plot.no function has some objectionable behavior [93] in understanding the whole modeling process. According to the ANN experts [94, 95] suggested that the modeling supply of the total weights is simple and valuable, as the interpretation of the model's predictor variable affects how the variable responds. This is due to the weighted limits of the information barriers that keep the learning outcomes. The general weights in the process were analyzed based on their closeness to 0 or below it; all of the generalized weights for the process predictor variables were typically above zero. The general weights distribution is demonstrated in Figs 3–6, which shows the predictor variables exerted on the non-linear result on the

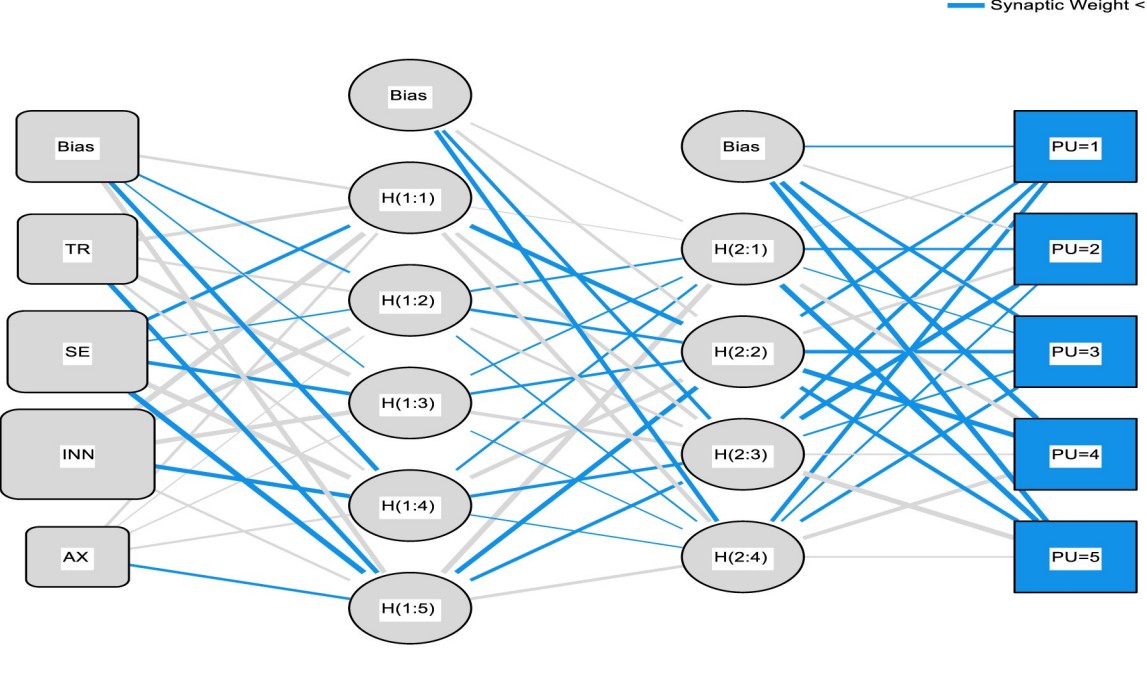

**Fig 3. ANN model (Part I).**

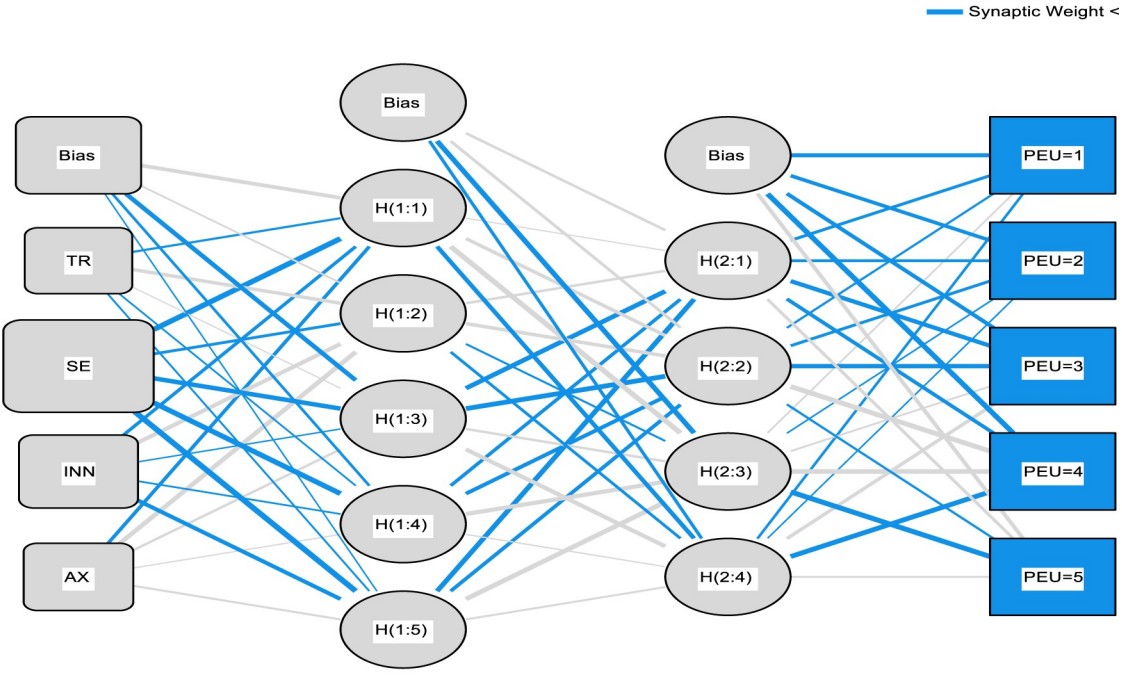

**Fig 4. ANN model (Part II).**

response variable. In a specific way, the overall impact of the electronic medical record in healthcare is non-linear (asymmetric). Cross-validation is another critical aspect of the modeling technique that helps build predictive models. Ten-fold cross-validation modeling with the resulting ratio of 70:30 data for testing and training the estimates was used to evade over-fitting. The mean square of error (MSE) from the model of 10 networks was used to examine the accuracy of the model.

## 4. Data analysis and findings

### 4.1 Personal/demographic data

The table below (Table 1) represents personal and demographic data obtained from the assessment which is as follows: 70 per cent of respondents were females with 30 per cent being males while 62 per cent of the respondents aged above 29 years and 38 per cent aged between 18 to 29 years. The level of education of the respondents was also considered with 65 per cent having a bachelor's degree, 23 per cent having a master's degree, 10 per cent were Ph.D. holders whereas the rest had diplomas.

### 4.2 A pilot study of the questionnaire

The questionnaire was initially prepared in the English language. After then, it was translated into the Arabic language. The translation and back-translation were performed by three native Arabic speakers, all of whom are fluent in Arabic and English languages. The questionnaire item's reliability was assessed in the pilot study. Thirty respondents from 15 different

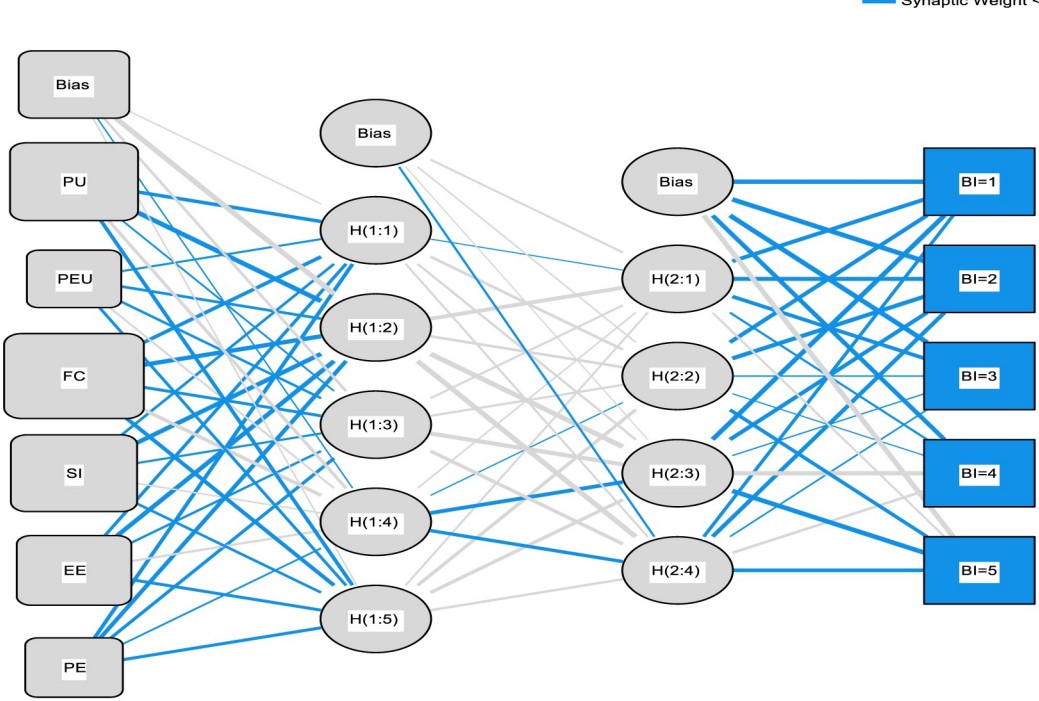

Fig 5. ANN model (Part III).

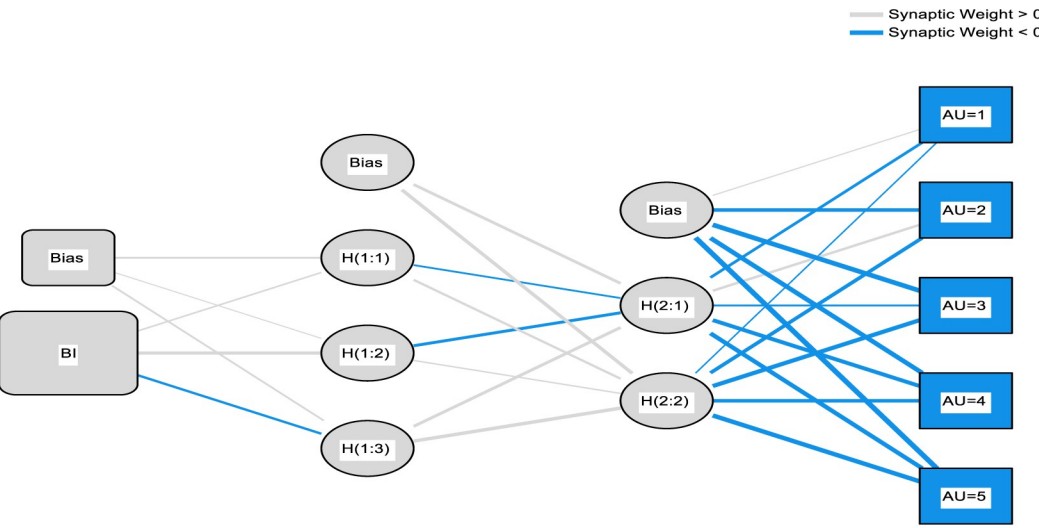

Fig 6. ANN model (Part IV).

**Table 1. Demographic information of participants.**

| Criterion | Factor | Frequency | Percentage |
|---|---|---|---|
| Gender | Female | 182 | 70% |
| | Male | 77 | 30% |
| Age | 18 to 29 | 98 | 38% |
| | 30 to 39 | 58 | 22% |
| | 40 to 49 | 54 | 21% |
| | 50 to 59 | 49 | 19% |
| Education qualification | Diploma | 5 | 2% |
| | Bachelor | 168 | 65% |
| | Master | 59 | 23% |
| | Doctorate | 27 | 10% |
| Experience | 1–5 | 52 | 20% |
| | 5–10 | 42 | 16% |
| | 10–15 | 60 | 23% |
| | 15–20 | 59 | 23% |
| | 20+ | 46 | 18% |
| Type of Sector | Federal / Government | 210 | 81% |
| | Private | 49 | 19% |

healthcare facilities in Dubai were selected randomly from a population to participate in the pilot study. The study's sample size was set at 300 respondents, depending on 10% of the total sample size for the research; the pilot study is contained in the main findings with the pilot participants being considered in the main research; however, the additional data is obtained from the respondents. The analysis of the obtained data was done using the Cronbach alpha test to determine the internal reliability of the pilot study; it reveals somehow satisfactory results from the measured items using the IBM SPSS statistics version 23. The reliability coefficient of 0.70 is seen as satisfactory when more attention is put into evaluating the studies [96]. The Cronbach alpha values are shown in Table 2 below for the five measurement scales listed.

## 4.3 Data analysis

The current study, unlike those conducted earlier, applied a hybrid analysis approach of both PLS-SEM and ANN methods to validate the formulated hypotheses regarding factors

**Table 2. Cronbach's Alpha values for the pilot study (Cronbach's Alpha ≥ 0.70).**

| Constructs | Cronbach's Alpha. |
|---|---|
| AU | 0.823 |
| AX | 0.803 |
| BI | 0.735 |
| EE | 0.842 |
| FC | 0.832 |
| INN | 0.851 |
| PEU | 0.790 |
| PU | 0.867 |
| PE | 0.778 |
| SE | 0.886 |
| SI | 0.820 |
| TR | 0.861 |

influencing the adoption of an EMR system. Firstly, the study analysed the research models using the PLS-SEM approach combined with the Smart-PLS software. This approach was preferred as the best in theoretical model research and due to the lack of past literature on the research topic. Also, experience from a similar line of research, the current study applied a two-step approach that is, a structural and measurement model in analysing the research model, and the IPMA which is an advanced PLS-SEM method used in identifying the significance and performance of the constructs present in this study. Furthermore, the constructs have been assessed in terms of multi-collinearity. The variance inflation factor (VIF) of the predictive variables was examined to see whether there was any difficulty with linearity.

The above analysis was then followed by the adoption of the ANN algorithm as a method of authentication, complement, and investigation of the PLS-SEM-analysis to confirm the independent variables' effectiveness on the dependent variable. The ANN algorithm is considered a function approximation tool effective in analysing relationships between input and output in complex or non-linear situations. It comprises three essential mechanisms; transfer function, learning rule, and network architecture which are further categorised into recurrent networks, feed-forward multilayer perception (MLP), and radian basis [97, 98]. Among the three, MLP is the most widely used approach with several inputs and outputs layers connected via hidden nodes significant for testing the proposed research model.

**4.3.1 Common method bias.** Common Method Bias (CMB) is one of the excellent wellsprings of estimating model blunder. As per [99], when the information is collected from one source at the moment for both endogenic and exogenic factors, CMB could exist. There are two fundamental ways to deal with moderating CMB: practical and arithmetical [99]. Harman's one-element examination is the most widely recognized geometric methodology that has been seen to test for CMB. The test shows hazardous CMB if an investigative component investigation (EFA) stacks all things from every one of the builds onto a solitary element, recommending the component represents a lot of divide fluctuation between the factors because of the method [100, 101]. On the off chance that not, the case is that CMB is anything but an unavoidable issue. Statistically, assuming that Harman's one component test results for a solitary variable record for half or more, there might be CMB issues in the data [102]. To test for CMB, Harman's one-factor test was applied. The outcomes show that complete scatterings are 38.63%, which fulfills the guideline of the half. This outcome shows that the current data are liberated from critical CMB issues.

**4.3.2 Multicollinearity assessment.** IBM SPSS was used to conduct a multicollinearity evaluation of trust, self-efficacy, personal innovativeness, and technology anxiety as the first set of constructs predicting the perceived ease of use and perceived usefulness level. The findings have shown that the predictor constructs (trust, self-efficacy, personal innovativeness, and technology anxiety) had VIF values less than 5 and a tolerance level higher than 0.2. Next, the multi-collinearity assessment of perceived ease of use and perceived usefulness was conducted being predictors of intention to use the EMR system in the future. Both constructs were found to be free of multicollinearity since they satisfied the VIF (less than 4.0) and tolerance level standards (greater than 0.2).

**4.3.3 Convergent validity.** In assessing the measurement model, it was essential to ascertain construct reliability consisting of Cronbach's alpha (CA), composite reliability (CR), and construct validity that includes convergent and discriminant validity. In measuring construct reliability, the table below (Table 3) presents values resulting from Cronbach's alpha with a range of 0.775 to 0.938 which according to [103] seem to have surpassed the set threshold of 0.7. However, the values from the table verify the construct reliability, hence an error-free construct. From the table, it is clear that the value of composite reliability is above the 0.7 mark suggested by [104], as it ranges between 0.735 and 0.940. In measuring convergent validity, it

**Table 3. Convergent validity results assuring acceptable values.**

| Construct Regarding the use of EMR | Items | Factor Loading | CA | CR | AVE | Sources |
|---|---|---|---|---|---|---|
| AU | AU I | .780 | 0.881 | 0.884 | 0.641 | [36, 70] |
|  | AU II | .720 |  |  |  |  |
|  | AU III | .849 |  |  |  |  |
| AX | AX I | .880 | 0.851 | 0.875 | 0.770 | [39] |
|  | AX II | .850 |  |  |  |  |
| BI | BI I | .783 | 0.784 | 0.863 | 0.600 | [71] |
|  | BI II | .731 |  |  |  |  |
|  | BI III | .892 |  |  |  |  |
| EE | EE I | .882 | 0.775 | 0.735 | 0.538 | [36, 71] |
|  | EE II | .849 |  |  |  |  |
|  | EE III | .849 |  |  |  |  |
| FC | FC I | .811 | 0.830 | 0.833 | 0.663 | [36, 71, 105] |
|  | FC II | .829 |  |  |  |  |
|  | FC III | .829 |  |  |  |  |
|  | FC IV | .831 |  |  |  |  |
| PI | INN I | .729 | 0.893 | 0.897 | 0.823 | [106, 107] |
|  | INN II | .829 |  |  |  |  |
|  | INN III | .731 |  |  |  |  |
| PEU | PEU I | .829 | 0.938 | 0.940 | 0.843 | [108] |
|  | PEU II | .829 |  |  |  |  |
|  | PEU III | .929 |  |  |  |  |
|  | PEU IV | .781 |  |  |  |  |
| PU | PU I | .759 | 0.928 | 0.929 | 0.874 | [108] |
|  | PU II | .733 |  |  |  |  |
|  | PU III | .729 |  |  |  |  |
| PE | PE I | .820 | 0.856 | 0.858 | .0.777 | [109] |
|  | PE II | .841 |  |  |  |  |
|  | PE III | .812 |  |  |  |  |
|  | PE IV | .861 |  |  |  |  |
| SE | SE I | .893 | 0.847 | 0.850 | 0.742 | [110] |
|  | SE II | .802 |  |  |  |  |
|  | SE III | .859 |  |  |  |  |
| SI | SI I | .829 | 0.813 | 0.899 | 0.621 | [109, 111] |
|  | SI II | .822 |  |  |  |  |
|  | SI III | .829 |  |  |  |  |
| TR | TR I | .861 | 0.820 | 0.822 | 0.735 | [112] |
|  | TR II | .720 |  |  |  |  |
|  | TR III | .830 |  |  |  |  |

is crucial to determine the extracted average variance (AVE) and the factor loading. From the table, the factor loading values are seen to exceed the standard 0.7 value, which is also the case with the extracted average variance ranging from 0.538 to 0.874, whereas the standard value is 0.5 that verifies the construct validity and in general satisfying the convergent validity requirements.

**4.3.4 Discriminant validity.** In determining the measurement of discriminant validity, the following criteria must be put into consideration [113]; Heterotrait-Monotrait ratio, and Fornell-Larker criterion. According to the table below (Table 4), the indicated square roots of

**Table 4. Fornell-Larcker scale.**

|  | AU | AX | BI | EE | FC | INN | PEU | PU | PE | SE | SI | TR |
|---|---|---|---|---|---|---|---|---|---|---|---|---|
| **AU** | **0.870** |  |  |  |  |  |  |  |  |  |  |  |
| **AX** | 0.113 | **0.857** |  |  |  |  |  |  |  |  |  |  |
| **BI** | 0.446 | 0.529 | **0.855** |  |  |  |  |  |  |  |  |  |
| **EE** | 0.535 | 0.602 | 0.430 | **0.819** |  |  |  |  |  |  |  |  |
| **FC** | 0.525 | 0.496 | 0.518 | 0.345 | **0.930** |  |  |  |  |  |  |  |
| **INN** | 0.606 | 0.652 | 0.417 | 0.379 | 0.587 | **0.817** |  |  |  |  |  |  |
| **PEU** | 0.257 | 0.562 | 0.402 | 0.494 | 0.520 | 0.402 | **0.891** |  |  |  |  |  |
| **PU** | 0.116 | 0.446 | 0.261 | 0.453 | 0.470 | 0.358 | 0.171 | **0.845** |  |  |  |  |
| **PE** | 0.159 | 0.425 | 0.268 | 0.427 | 0.403 | 0.257 | 0.131 | 0.233 | **0.859** |  |  |  |
| **SE** | 0.136 | 0.462 | 0.336 | 0.575 | 0.467 | 0.328 | 0.086 | 0.297 | 0.136 | **0.862** |  |  |
| **SI** | 0.500 | 0.645 | 0.602 | 0.360 | 0.729 | 0.422 | 0.139 | 0.576 | 0.500 | 0.645 | **0.802** |  |
| **TR** | 0.112 | 0.689 | 0.342 | 0.508 | 0.434 | 0.404 | 0.156 | 0.444 | 0.112 | 0.689 | 0.342 | **0.808** |

the extracted average variance values are higher than the correlational constructs hence the Fornell-Larker criterion being acceptable [114].

According to Table 5, the values representing the Heterotrait-Monotrait ratio seem to be below [115] standard threshold of 0.85 hence affirming the Heterotrait-Monotrait ratio which verifies the discriminant validity. Therefore, it is precise that the study's validity and reliability while assessing the measurement model are free from errors hence the collected data being efficient for the research.

**4.3.5 Model fit.** Table 6 was used to measure PLS-SEM goodness-of-fit for the study. With the study's value of RMS theta being 0.066, the PLS model is proved to be valid. According to [113], any value representing RMS theta ranging between 0.00 and 0.12 is considered to be a good fit for the PLS-SEM model.

## 4.4 Hypotheses testing

The following study proposed the use of a complementary approach in testing hypotheses hence ANN algorithms and PLS-SEM model were utilised as such an approach is believed to enhance information system literature which in this case is predicting the intention of using an EMR system in healthcare. According to [116], the PLS-SEM model is significant when

**Table 5. Heterotrait-Monotrait ratio (HTMT).**

|  | AU | AX | BI | EE | FC | INN | PEU | PU | PE | SE | SI | TR |
|---|---|---|---|---|---|---|---|---|---|---|---|---|
| **AU** |  |  |  |  |  |  |  |  |  |  |  |  |
| **AX** | 0.209 |  |  |  |  |  |  |  |  |  |  |  |
| **BI** | 0.087 | 0.204 |  |  |  |  |  |  |  |  |  |  |
| **EE** | 0.095 | 0.214 | 0.107 |  |  |  |  |  |  |  |  |  |
| **FC** | 0.316 | 0.492 | 0.565 | 0.268 |  |  |  |  |  |  |  |  |
| **INN** | 0.423 | 0.617 | 0.613 | 0.330 | 0.675 |  |  |  |  |  |  |  |
| **PEU** | 0.502 | 0.496 | 0.612 | 0.309 | 0.561 | 0.730 |  |  |  |  |  |  |
| **PU** | 0.465 | 0.617 | 0.691 | 0.378 | 0.650 | 0.791 | 0.116 |  |  |  |  |  |
| **PE** | 0.493 | 0.646 | 0.677 | 0.369 | 0.667 | 0.728 | 0.081 | 0.635 |  |  |  |  |
| **SE** | 0.072 | 0.469 | 0.260 | 0.351 | 0.491 | 0.274 | 0.045 | 0.255 | 0.624 |  |  |  |
| **SI** | 0.240 | 0.396 | 0.267 | 0.419 | 0.447 | 0.280 | 0.023 | 0.236 | 0.409 | 0.240 |  |  |
| **TR** | 0.259 | 0.551 | 0.405 | 0.406 | 0.522 | 0.426 | 0.111 | 0.355 | 0.595 | 0.259 | 0.551 |  |

**Table 6. Model fit indicators.**

| | Complete Model | |
|---|---|---|
| | **Saturated Model** | **Estimated Model** |
| **SRMR** | 0.031 | 0.032 |
| **d_ULS** | 0.771 | 1.312 |
| **d_G** | 0.567 | 0.570 |
| **Chi-Square** | 440.225 | 445.681 |
| **NFI** | 0.836 | 0.840 |
| **RMS Theta** | **0.066** | |

**Note:** SRMR- Standard root mean square residual, dULS- Squared Euclidean Distance, dG- Geodesic Distance, NFI-Normal Fit Index, RMS_theta- root mean squared residual covariance matrix of the outer model residuals (Dijkstra & Henseler, 2015; Lohmöller, 1989).

validating a conceptual model, and predicting dependent variables based on existing theory. On the other hand, the ANN algorithm is effective when finding a dependent variable using independent variables.

**4.4.1 Hypotheses testing using PLS-SEM.** The table above (Table 7) represents the path analysis using *p*-values, *t*-values, and path coefficients which shows with a precision that the study's hypotheses are valid as they are supported by empirical data. According to [117], values in the path coefficient above 0.67, values between 0.33 and 0.67, and 0.19 and 0.33 are considered as high, moderate, and weak respectively with any value less than 0.19 being inadmissible for analysis. According to results from the table, perceived usefulness is proved to have a positive relationship with technology anxiety, trust, self-efficacy, and personal innovativeness hence hypotheses 1a, 2a, 3a, and 4a are validated respectively. Similarly, PEU is seen to be influenced by trust, self-efficacy, personal innovativeness, and technology anxiety hence hypotheses 1b, 2b, 3b, and 4b respectively being admissible. Further results on BI regarding the use of an EMR system show a significant influence by PU, PEU, PE, EE, SI, and FC hence validating hypotheses 5, 6, 7, 8, 9, and 10 respectively. Also, results showing the relationship

**Table 7. Hypotheses-testing of the research model.**

| Hypotheses (H) | Relationship | Path | *t*-value | *p*-value | Direction | Decision |
|---|---|---|---|---|---|---|
| 1a | TR-> PU | 0.777 | 12.104 | 0.000 | + | Accepted |
| 1b | TR -> PEU | 0.337 | 3.717 | 0.026 | + | Accepted |
| 2a | SE-> PU | 0.442 | 2.208 | 0.032 | + | Accepted |
| 2b | SE -> PEU | 0.640 | 12.362 | 0.000 | + | Accepted |
| 3a | INN-> PU | 0.483 | 14.105 | 0.000 | + | Accepted |
| 3b | INN -> PEU | 0.551 | 10.248 | 0.000 | + | Accepted |
| 4a | AX-> PU | 0.306 | 2.765 | 0.029 | + | Accepted |
| 4b | AX -> PEU | 0.440 | 15.576 | 0.001 | + | Accepted |
| 5 | PU-> BI | 0.379 | 14.589 | 0.000 | + | Accepted |
| 6 | PEU-> BI | 0.313 | 2.633 | 0.023 | + | Accepted |
| 7 | PE-> BI | 0.424 | 2.066 | 0.041 | + | Accepted |
| 8 | EE-> BI | 0.514 | 3.925 | 0.038 | + | Accepted |
| 9 | SI-> BI | 0.281 | 2.067 | 0.033 | + | Accepted |
| 10 | FC-> BI | 0.653 | 9.687 | 0.000 | + | Accepted |
| 11 | BI-> AU | 0.748 | 13.128 | 0.000 | + | Accepted |

**Note:** + (positive)

Table 8. R$^2$ of the endogenous latent variables.

| Constructs | R$^2$ | Predictive power |
|---|---|---|
| AU | 0.736 | High |
| BI | 0.753 | High |
| PEU | 0.693 | High |
| PU | 0.721 | High |

between BI and the actual use of an EMR system proved to be statistically significant hence hypothesis 11 being valid for the study.

From Fig 2 and Table 8 above, the utilised model is proved to have a significantly high predictive power in support of the variance regarding the use of an EMR system.

**4.4.2 Artificial Neural Network results.** The second analysis of factors related to the use of EMR is the ANN algorithm which consists of one input neuron which is the actual use of an EMR system, and several output neurons. However, the ANN algorithm can only be used to analyse factors (output neurons) that are proved valid by the PLS-SEM model which in this case are; social influence, self-efficacy, performance expectancy, perceived usefulness, perceived ease of use, personal innovativeness, facilitating conditions, effort expectancy, behavioural intention, and technology anxiety. Further, the accuracy of the ANN approach is proved significant due to there being a very small difference of 0.0086 and 0.0039 between the study's calculated root mean square of error and the standard deviation of both, testing and training data respectively. Also, as a way to enhance the model's performance, a standardised range of o and 1 was applied for both output and input neurons [56], and a cross-validation ratio of 70:30 for both training and testing data respectively to prevent over-fitting while using the ANN model [97]. The ANN analysis is further demonstrated by Figs 3–6 below.

**4.4.3 Sensitivity analysis.** Table 9 above presents the normalised importance of the study's Artificial Neural Network EMR-predictors from the highest to the lowest in percentage that is computed using the average value of each predictor against their highest mean value. Also, from the analysis in both the PLS-SEM model and the ANN model, the predictive power of the latter is found to be more accurate, hence the above results being trustworthy. The accuracy of the two models was tested using [118] goodness of fit estimation, whereby ANN's model scored 89% compared to PLS-SEM's model with a score of 73.6%. The difference in accuracy can be attributed to ANN's approach of deep learning used to determine the non-linear linkages between the constructs.

Table 9. Independent variable importance.

| | Mean Importance | Normalised Importance |
|---|---|---|
| BI | .459 | 100.0% |
| INN | .390 | 95.0% |
| FC | .375 | 93.4% |
| PU | .220 | 79.8% |
| SE | .319 | 77.8% |
| SI | .212 | 76.9% |
| EE | .156 | 56.8% |
| TR | .189 | 46.2% |
| PE | .076 | 27.5% |
| PEU | .061 | 22.2% |
| AX | .083 | 20.1% |

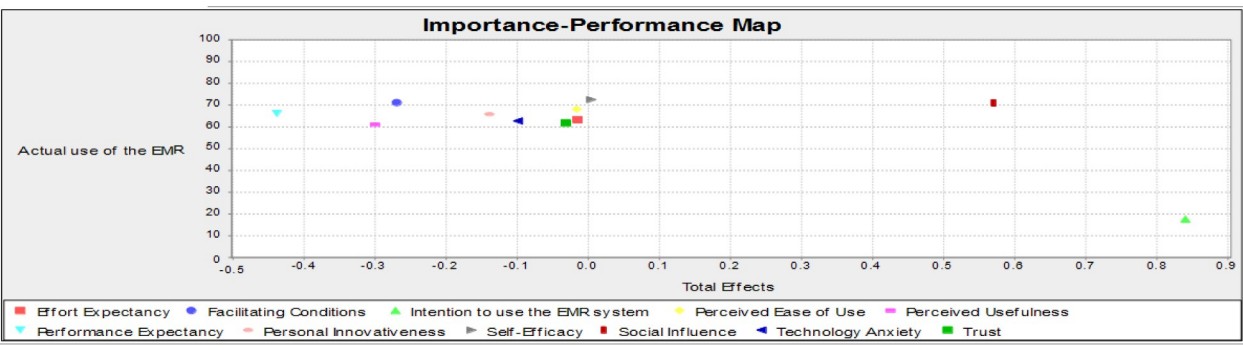

**Fig 7. IPMA results.**

**4.4.4 Importance-Performance Map Analysis.** The findings were further elaborated for management implications using Importance-Performance Map Analysis (IPMA) [119]. The IPMA findings are divided into performance and significance [119, 120]. Performance is normally graded on a scale of 0 to 100. According to [119], Importance-Performance Map Analysis is an improved technique used to analyse the results of the PLS-SEM approach [121]. The Importance-Performance Map Analysis (IPMA) involves the use of a target variable for the analysis, which in this case is the actual use of an EMR system. Fig 7 shows the IPMA results for the study's factors influencing the use of EMR. According to the analysis, the behavioural intention has the highest score in both performance and importance measures, while self-efficacy and performance expectancy has the lowest scores in performance measures and importance measures respectively.

The relative relevance results from PLS-SEM, ANN, and IPMA are summarized in Table 10. The results of the PLS-SEM and IPMA analyses show the relative importance ranking of the predictive variables "performance expectancy, facilitating conditions of use, honesty, personal creativeness, consciousness, new tech anxiety, system quality, user satisfaction, subjective norms, facilitating conditions, and the intentions to use the EMR system." However, according to the ANN model, "personal creativeness, relative advantage, performance expectancy, consciousness, perceived trust, reliability, user satisfaction, effort expectancy, and facilitating conditions of use, technology anxiety" are the most important predictors of actual EMR use.

**Table 10. Summary of ranking importance.**

| Output: Actual use of the EMR | PLS-SEM | IPMA | ANN sensitively |
|---|---|---|---|
| BI | 1 | 1 | 1 |
| INN | 7 | 7 | 2 |
| FC | 8 | 8 | 3 |
| PU | 9 | 9 | 4 |
| SE | 3 | 3 | 5 |
| SI | 2 | 2 | 6 |
| TR | 5 | 5 | 7 |
| EE | 4 | 4 | 8 |
| PE | 10 | 10 | 9 |
| PEU | 4 | 4 | 10 |
| AX | 6 | 6 | 11 |

## 5. Discussion and conclusion

The aim of this empirical study was to investigate factors that predict and influence the use and acceptance of an EMR system in the healthcare system. To archive this, we employed an integrated approach using UTAUT and TAM constructs and integrated them with the TAM external factors. This further revealed a significant effect instilled by UTAUT and TAM factors towards BI regarding the use of an EMR system capable of influencing the adoption with four of the external factors proving critical in making the process a success.

According to the current research, it can be proved that PU and PEU are significantly and positively affected by the TAM external aspects, hence influential in the adoption of an EMR system. Study [36] proves that trust factors play an influential role in PU and PEU regarding the acceptance of health technology in developed nations. The same is further proved by [38], who state that the link between trust factors and PU is significant in a user's decision in accepting new technology. Moreover, trust beliefs regarding PEU argue that trust can reduce a technology user's need to control, monitor, and understand the situation, hence reducing the cognitive effort further enhancing transactions and digital communications. Similarly, [37] believe that an efficient level of trust is capable of highly influencing PEU.

Another factor influencing PU and PEU is self-efficacy, whereby research shows that a high self-efficacy level is related to a positive effect compared to a lower level of the same. According to [23, 41], self-efficacy has a direct influence on PU and PEU in accepting healthcare technology. The same is also applicable in personal innovativeness, whereby this factor is proved to heighten the level of confidence towards a user's ability. [42] also posit that the higher the level of innovativeness in a person, the higher they are likely to have a high technology perception. In addition to this is [6, 42, 44] findings which, alongside the TAM theory, prove that PU and PEU are directly linked to personal innovativeness. [45–47], on the contrary, prove a significant effect towards PU and PEU by technology anxiety in the adoption of healthcare technology.

Research previously conducted in the medical field tends to agree with this current research in the fact that healthcare providers are willing and ready to use technology if it proves to be useful and easy to use. Such a study is by [48], who facilitated the traditional adoptive theory with the notion that PU and PEU play a significant role in determining the adoption of healthcare technology. Further findings from [4] demonstrate that PU and PEU are capable of instilling a positive attitude towards BI and the actual use of an EMR system. Also in line with the current study is [52, 55, 56] study, whose findings proves that PE is one of the highly influential factors towards BI of an individual in using new technology to facilitate health information systems at large.

Furthermore, studies by [58, 122] imply that EE has a positive impact on a user's behavioural intention towards the use of eHealth services, M-health services, and EMR systems. Behavioural influence is further proved to be influenced by FC and SI regarding the acceptance of new information technology in the healthcare setting as seen in [60, 61, 123] studies. The above is also echoed by [2], who proves the existence of a strong relationship between SI and BI regarding the adoption of health information systems.

In conclusion, this current research proves that behavioural intention is directly influential of a user's actual use of an EMR system, which is supported by [124]. Also, according to [34, 63, 65], behavioural intention acts as a predictor regarding the actual use of new technology by an individual and an essential beginner step in the adoption of a technological system, and more specifically an electronic medical record system in the healthcare setting [2, 17, 66].

### 5.1 Implication and limitation

**5.1.1 Practical/managerial implications.**    This study adds relevance to the research concerned with electronic medical records in the medical field, with more significance in the UAE

and other developed countries. Theoretically, this study proves the importance of personal innovativeness, facilitating conditions, and behavioural intention-paths of influence towards the adoption of an EMR system by healthcare providers. Besides, UTAUT and TAM factors utilised in the examination of healthcare practitioners' behavioural intention towards the use of an EMR system prove to be highly effective predictive and mediating mechanisms in influencing the use and acceptance of an EMR system by the healthcare providers. It is however notable that previous research on eHealth is majorly conducted based on the developed nation's context [2, 16] hence the need for more studies based on a different context. Further, according to the review of literature on current research regarding healthcare technology, UTAUT and TAM models have been employed to determine factors that predict the use and acceptance of an EMR system. However, researchers have rarely utilised a comprehensive and hybrid technology model that is capable of covering distinctive trends on a wide scope in healthcare. Therefore, this study proves to be significant in filling such gaps through hypotheses testing and the further exploration of UTAUT and TAM models.

In essence, this study's main is determining effects brought about by preceding factors influencing the adoption of an EMR system both directly and indirectly. Such an investigation would, however, require an effective implementation with the use of an integrated and innovative research model as enhancing factors to assist in understanding the determinants of the adoption of an EMR system. It, therefore, is notable that the current study's development of a conceptual model combining both UTAUT theory and TAM model with identified external factors in highlighting the predictableness and significance of the findings. This research has further put into use various external variables that are precise to the significance of EMR's external factors in the healthcare setting. It is notable that this study has uniquely applied external factors that are different from past literature which focused on confidentiality, usability, and availability. In addition, this study employed a new methodology regarding the analysis of collected data with the use of the ANN and PLS-SEM approaches that are also recognised to be among the best tools of analysis in the prediction of health technology adoption. Also, as far as reviewed literature on the adoption of EMR is concerned, this study is termed as the first to attempt the use of the PLS-SEM and ANN approach together with an integrated model (UTAUT & TAM) approach in an aim to fill the gap on the literature about this study's topic. Furthermore, this study's findings prove to have significant implications for health information officers, government agents, and hospital managers as they play a critical role with dominant power in the implementation of EMR systems in healthcare facilities. Hospital managers are also aware of individual features presented by various healthcare providers which might be influential to the acceptance of an EMR system and its successful adoption.

It is worth noting that an effective EMR in the provision of quality service is supposed to take into consideration factors such as innovativeness, facilitating conditions, and the desired stability of EMR systems to have the capability of responding to the requirements of medical practitioners in providing healthcare services. Additionally, it is essential to gain knowledge in understanding the impact that individual aspects of healthcare providers have on EMR and its successful implementation that can further contribute towards the provision of better quality healthcare services.

**5.1.2 Theoretical implications.** In terms of methodology, this work, unlike previous empirical investigations that relied solely on SEM analysis, employs a hybrid SEM-ANN strategy based on deep learning to contribute to the literature in general and the healthcare domain in particular. The ANN model outperforms the PLS-SEM model in terms of predictive capability. We infer that ANN analyses' increased predictive power is due to the deep ANN architecture's ability to uncover non-linear correlations between the elements in the theoretical model.

**5.1.3 Limitations and future research directions.** This study, however, experienced limitations that should be put into consideration to avoid them while conducting similar research in the future. Although we focused on specific external factors aimed at enhancing EMR's visibility, it is recommendable for further research on the same topic to utilise external features that conform to new technological developments and the use of EMR. Also, with this study being limited to having its focus on particularly UTAUT and TAM models, some important aspects in the social and psychological areas might have been left out. Also, the current study utilised a cross-sectional study design in the collection of its data which was over a short period, hence the process not being extensive and thorough hence the need to consider similar research with the incorporation of longitudinal study designs with the provision of ample time for more precise results. Finally, the current study lacked diversity while collecting data as it only utilised questionnaires as its only data collecting tool hence recommending the use of different tools of data collection that would make the findings extensive.

This research might be expanded to incorporate more medical treatments in the future, and further simulation studies could be undertaken in other developing nations with a different design, demographic, or context. The research may be enlarged by revamping and automating the patient transition between triage and treatment rooms. The sharing of clinical data and the integration of different healthcare information systems among two or more healthcare institutions is a vacuum in the literature that needs to be addressed.

## Author Contributions

**Conceptualization:** Ahmad Aburayya.

**Data curation:** Ahmad Aburayya.

**Investigation:** Amina Almarzouqi.

**Methodology:** Said A. Salloum.

**Project administration:** Amina Almarzouqi.

**Software:** Said A. Salloum.

**Supervision:** Amina Almarzouqi.

**Validation:** Said A. Salloum.

**Visualization:** Said A. Salloum.

**Writing – original draft:** Ahmad Aburayya.

**Writing – review & editing:** Said A. Salloum.

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
