## [Decision Letter · Decision Letter 0]

19 Apr 2022

PONE-D-22-05312Determinants Predicting the Electronic Medical Record Adoption in Healthcare: A SEM- Artificial Neural Network ApproachPLOS ONE

Dear Dr. Salloum,

Thank you for submitting your manuscript to PLOS ONE. After careful consideration, we feel that it has merit but does not fully meet PLOS ONE’s publication criteria as it currently stands. Therefore, we invite you to submit a revised version of the manuscript that addresses the points raised during the review process.

We look forward to receiving your revised manuscript.

Kind regards,

Anandakumar Haldorai, PhD

Academic Editor

PLOS ONE

Journal Requirements:

3. Please ensure that you refer to Figure 7 in your text as, if accepted, production will need this reference to link the reader to the figure.

Additional Editor Comments:

Please carefully address the issues raised in the comments and, up front in your revised paper. Your revised paper will be sent to the same reviewers, as well as possibly new reviewers, for evaluation.

Make sure the Abstract briefly describes the paper as it is used in abstracting and citation services. Clearly specify the Purpose, Methodology, and problem findings.

Include a list of six to ten key words after the Abstract.

Spell out each acronym the first time used in the body of the paper. Spell out acronyms in the Abstract only if used there.

You may ignore any suggestion of including self-references by reviewers if not applicable.

Include a paragraph at the end of the Introduction describing the organization of the paper.

Make sure that the Conclusion briefly summarizes the results of the paper it should not repeat phrases from the Introduction. Keep the Conclusion to about 200 words. Do not use any references or acronyms in the Conclusion.

Make sure all figures and tables are referred to in the body of the paper.

It is recommended to use a professional proofread and native English correction. Papers with less than excellent English will not be published even if technically perfect.

Reviewers' comments:

Reviewer's Responses to Questions

**Comments to the Author**

1. Is the manuscript technically sound, and do the data support the conclusions?

Reviewer #1: Yes

Reviewer #2: Partly

Reviewer #3: Yes

2. Has the statistical analysis been performed appropriately and rigorously? 

Reviewer #1: Yes

Reviewer #2: Yes

Reviewer #3: Yes

3. Have the authors made all data underlying the findings in their manuscript fully available?

Reviewer #1: No

Reviewer #2: Yes

Reviewer #3: Yes

4. Is the manuscript presented in an intelligible fashion and written in standard English?

Reviewer #1: Yes

Reviewer #2: Yes

Reviewer #3: Yes

5. Review Comments to the Author

Reviewer #1: This paper investigates factors that predict and influence the use and acceptance of an EMR system in the healthcare system in the UAE using SEM and ANN. In general, the manuscript is well written and very comprehensive. Although I highly recommend authors consider submitting the paper to PLoS Digital Health instead since it perfectly fits the scope of the journal, I would recommend it to this journal subject to some revision:

The last two sentences in the “Abstract” seem ‘not necessary’. The first one says, “In addition, the study offers methodological significance as it proves the efficiency of the ANN algorithm in determining non-linear linkages”, such a conclusion is a bit beyond the scope of the paper; after all, this is not a ‘pure’ machine learning study. As for the second one “It can also be concluded that the adoption of an EMR system is in high demand in the healthcare setting as it enhances the transmission and sharing of information within and between healthcare institutions.”, I would say this is a ‘trivial’ conclusion.

Is “Literature Review” mentioned in line 117 a convenient title to the successive paragraphs? I am not able to see the ‘border’ between the section entitled “Introduction” and the other one entitled “Literature Review”. I would remove that title and keep them all under “Introduction”. In the same context, lines 118-125 contain nothing, but a redundancy of information already discussed in the “Introduction”; I would remove them as well.

• Confusion between developing and developed countries:

What is surprising in this manuscript is the implicit mention that UAE is a developing country!!! For example, it is mentioned in Line 592 “with more significance in the UAE and other developing countries” ... I ask the authors to clarify this point, I would definitely disagree to classify UAE as a developing country, it is one of the most developed countries in the world.

In line 79, after talking about developing countries, authors suddenly moved to discuss the use of EMR in the USA! The same occurs in some other instances in the manuscript.

Incentives, challenges, effects… of adopting EMR in developed countries should be discussed separately from those in developing countries. Well, I understand that many factors could be common, but that confusion should be addressed. In other words, authors are asked to explain explicitly and plainly the problem of the EMR overall (common factors amongst all countries), then to differentiate between the scope of the problem in developing and developed countries and clarify the case of UAE.

Have the questionnaires used in this paper been used somewhere else? I mean in other particle?

• Acronyms:

Some abbreviations are used – some of them too many times - without explanation in their first instances:

UTAUT is never explained as Unified Theory of Acceptance and Use of Technology. Speaking of which, I was wondering if the terms mentioned between brackets in line 173 as “(user behaviour and acceptance analysis)” next to UTAUT meant to be a description of some the model’s characteristics. I believe it is confusing a bit.

TAM is never explained as Technology Acceptance Model.

ANN is explained as Artificial Neural Networks in line 339, not in the first instance it was used.

SEM is never explained as Structural Equation Modeling

ICT is never explained as Information and Communications Technology

NHIN is never explained as Nationwide Health Information Network

PLS-SEM is never explained as Partial Least Squares Structural Equation Modeling

IPMA is explained as Importance-Performance Map Analysis in line 529, not in the first instance it was used.

And BI…

Please make sure you pass through all acronyms and double-check.

• Minor mistakes/comments:

In line 86, I think there is something missing in the English… “there is research handling”

In line 94, what research in “the research”? the current one? If not, what reference?

One expects a reference in line 113, after “technology”. Same applies to line 115 after “study” and line 140 after “data”.

In line 173, “The researcher” is not well fitted in the sentence.

In lines 178-179: I think capital letters are not necessary here.

In line 190, a comma is to be inserted after (37).

In line 225, period to be removed after “connection”.

In table 1, I think “frequency” is mentioned twice as a column header wrongly.

In Table 5, “Mod” to be written as “Model”

In line 545, who is “the researcher”? same question in line 637.

In line 552, I suppose something is wrong “Aligning with this is (37) study proves”

In line 607, I suppose there is a missing word after “main”.

Reviewer #2: Dear authors,

Thank you for the opportunity to review your work. The paper explores an interesting topic in general, but lacks clarity in its novelty and requires enrichment of the research idea and improvement of its methodology/analytical and theoretical base.

1. In the abstract the author(s) wrote “SEM-PLS”, you probably mean PLS-SEM which stands for partial least squares structural equation modeling. Please address this issue where necessary.

2. There is a great need to highlight the aim, gap, importance and contributions of this study in the intro section. Based on my readings, the section appears as pile up information without clear bearings. Please try to develop the section to enhance clarity.

3. In page 13, line 337. The authors stated that they will use PLS-SEM and ANN. But failed to provide detailed explanation about the pros and cons of the two methods. Below are two valuable studies that you can find details about the usage of the methods. Read and integrate the information available in the papers below:

Abubakar, A. M., Behravesh, E., Rezapouraghdam, H., & Yildiz, S. B. (2019). Applying artificial intelligence technique to predict knowledge hiding behavior. International Journal of Information Management, 49, 45-57.

Yakubu, M. N., Dasuki, S. I., Abubakar, A. M., & Kah, M. M. (2020). Determinants of learning management systems adoption in Nigeria: A hybrid SEM and artificial neural network approach. Education and Information Technologies, 25(5), 3515-3539.

4. Information about the language of study, back-translation and pilot survey were not provided by the authors. The left the reviewer wondering, I suggest adding such information with reference and evidences.

5. The authors said they used a cross-sectional and self-rated surveys but did not gauge for common method bias. Why? This has to be done. Also, the authors did not control for non-response bias. Why?

6. I commend the authors approach to reliability, convergent and discriminant validity. Well done. However, the study has several predictors which makes the model vulnerable to multi-collinearity issue. Please consider a test to rule out this problem.

7. The study reported the beta, t-value, p-value and R-squared, all of which I commend. However, the f-squared values were withheld. Any reason for this, reporting the values will be great.

8. ANN analyses are well conducted, except my previous comments about the pros and cons, which you can find in the suggested paper.

9. The implication is too abstract, I am unable to follow nor understand the theoretical or practical implications. Please develop a sub-section and their respective contents as follows:

-Theoretical implications

-Practical/Managerial implications

-Limitations and future research directions

I wish the authors all the best.

Reviewer #3: Overall, the study is conducted well. However, I have a few concerns.

What is the research question?

The results didn't provide the ranking of the factors in terms of their significance, See the following references which is also missing in the paper.

Sohaib, O., Hussain, W., Asif, M., Ahmad, M. and Mazzara, M., 2019. A PLS-SEM neural network approach for understanding cryptocurrency adoption. IEEE Access, 8, pp.13138-13150.

Alharbi, A., & Sohaib, O. (2021). Technology readiness and cryptocurrency adoption: PLS-SEM and deep learning neural network analysis. IEEE Access, 9, 21388-21394.

6. PLOS authors have the option to publish the peer review history of their article (what does this mean?). If published, this will include your full peer review and any attached files.

Reviewer #1: No

Reviewer #2: No

Reviewer #3: No

---

## [Author Response · Author response to Decision Letter 0]

9 Jun 2022

Reviewer 1: I have incorporated all of your suggestions into my revision. They were very helpful. Thank you.

Reviewer 2: I have incorporated all of your suggestions into my revision. Thank you for your help.

Reviewer 3: I have incorporated all of your suggestions into my revision. They were very helpful. Thank you.

---

## [Decision Letter · Decision Letter 1]

28 Jun 2022

PONE-D-22-05312R1Determinants Predicting the Electronic Medical Record Adoption in Healthcare: A SEM- Artificial Neural Network ApproachPLOS ONE

Dear Dr. Salloum,

Thank you for submitting your manuscript to PLOS ONE. After careful consideration, we feel that it has merit but does not fully meet PLOS ONE’s publication criteria as it currently stands. Therefore, we invite you to submit a revised version of the manuscript that addresses the points raised during the review process.

We look forward to receiving your revised manuscript.

Kind regards,

Anandakumar Haldorai, PhD

Academic Editor

PLOS ONE

Journal Requirements:

Reviewers' comments:

Reviewer's Responses to Questions

**Comments to the Author**

1. If the authors have adequately addressed your comments raised in a previous round of review and you feel that this manuscript is now acceptable for publication, you may indicate that here to bypass the “Comments to the Author” section, enter your conflict of interest statement in the “Confidential to Editor” section, and submit your "Accept" recommendation.

Reviewer #1: All comments have been addressed

Reviewer #3: (No Response)

2. Is the manuscript technically sound, and do the data support the conclusions?

Reviewer #1: Yes

Reviewer #3: Yes

3. Has the statistical analysis been performed appropriately and rigorously? 

Reviewer #1: Yes

Reviewer #3: Yes

4. Have the authors made all data underlying the findings in their manuscript fully available?

Reviewer #1: Yes

Reviewer #3: (No Response)

5. Is the manuscript presented in an intelligible fashion and written in standard English?

Reviewer #1: Yes

Reviewer #3: (No Response)

6. Review Comments to the Author

Reviewer #1: I would like to thank the authors for addressing my comments convincingly and adequately. I have no further notes to mention.

Reviewer #3: The authors mentioned in the response that the suggested references are included for IPMA. But, I didn't find the references as it's not included.

7. PLOS authors have the option to publish the peer review history of their article (what does this mean?). If published, this will include your full peer review and any attached files.

Reviewer #1: No

Reviewer #3: No

---

## [Author Response · Author response to Decision Letter 1]

29 Jun 2022

Reviewer 1: I have incorporated all of your suggestions into my revision. They were very helpful. Thank you.

Reviewer 2: I have incorporated all of your suggestions into my revision. Thank you for your help.

Reviewer 3: I have incorporated all of your suggestions into my revision . They were very helpful. Thank you.

---

## [Decision Letter · Decision Letter 2]

26 Jul 2022

Determinants Predicting the Electronic Medical Record Adoption in Healthcare: A SEM- Artificial Neural Network Approach

PONE-D-22-05312R2

Dear Dr. Salloum,

We’re pleased to inform you that your manuscript has been judged scientifically suitable for publication and will be formally accepted for publication once it meets all outstanding technical requirements.

Kind regards,

Anandakumar Haldorai, PhD

Academic Editor

PLOS ONE

Additional Editor Comments (optional):

Recommended for publication.

Reviewers' comments:

Reviewer's Responses to Questions

**Comments to the Author**

1. If the authors have adequately addressed your comments raised in a previous round of review and you feel that this manuscript is now acceptable for publication, you may indicate that here to bypass the “Comments to the Author” section, enter your conflict of interest statement in the “Confidential to Editor” section, and submit your "Accept" recommendation.

Reviewer #1: All comments have been addressed

Reviewer #3: All comments have been addressed

2. Is the manuscript technically sound, and do the data support the conclusions?

Reviewer #1: Yes

Reviewer #3: Yes

3. Has the statistical analysis been performed appropriately and rigorously? 

Reviewer #1: Yes

Reviewer #3: Yes

4. Have the authors made all data underlying the findings in their manuscript fully available?

Reviewer #1: Yes

Reviewer #3: Yes

5. Is the manuscript presented in an intelligible fashion and written in standard English?

Reviewer #1: Yes

Reviewer #3: Yes

6. Review Comments to the Author

Reviewer #1: My comments have already been addressed in the first round. I have no further comments/corrections/concerns.

Reviewer #3: All my previous comments addressed in the version. I don't have any further comments. All the best!!

7. PLOS authors have the option to publish the peer review history of their article (what does this mean?). If published, this will include your full peer review and any attached files.

Reviewer #1: No

Reviewer #3: No

---

## [Editor Report · Acceptance letter]

1 Aug 2022

PONE-D-22-05312R2 

Determinants Predicting the Electronic Medical Record Adoption in Healthcare: A SEM- Artificial Neural Network Approach 

Dear Dr. Salloum:

I'm pleased to inform you that your manuscript has been deemed suitable for publication in PLOS ONE. Congratulations! Your manuscript is now with our production department. 

Kind regards, 

on behalf of

Dr. Anandakumar Haldorai 

Academic Editor

PLOS ONE